# SS18 regulates pluripotent-somatic transition through phase separation

Junqi Kuang[1,2,3,9], Ziwei Zhai[1,2,3,9], Pengli Li[1,2,3], Ruona Shi[1,2,3], Wenjing Guo[1], Yuxiang Yao[4], Jing Guo[1,2], Guoqing Zhao[5], Jiangpin He[1,2], Shuyang Xu[1,2], Chuman Wu[1,2], Shengyong Yu[1,2,3], Chunhua Zhou[1,2,3], Linlin Wu[6], Yue Qin[1,2,3], Baomei Cai[1,2], Wei Li[1,2,3], Zichao Wu[1,2,5], Xiaoxi Li[1,2,5], Shilong Chu[1,2], Tingting Yang[1,2,3], Bo Wang[1,2], Shangtao Cao[1,2], Dongwei Li[1,2], Xiaofei Zhang[1,2], Jiekai Chen [1,2], Jing Liu [1,2,7✉] & Duanqing Pei [8✉]

The transition from pluripotent to somatic states marks a critical event in mammalian development, but remains largely unresolved. Here we report the identification of SS18 as a regulator for pluripotent to somatic transition or PST by CRISPR-based whole genome screens. Mechanistically, SS18 forms microscopic condensates in nuclei through a C-terminal intrinsically disordered region (IDR) rich in tyrosine, which, once mutated, no longer form condensates nor rescue SS18$^{-/-}$ defect in PST. Yet, the IDR alone is not sufficient to rescue the defect even though it can form condensates indistinguishable from the wild type protein. We further show that its N-terminal 70aa is required for PST by interacting with the Brg/Brahma-associated factor (BAF) complex, and remains functional even swapped onto unrelated IDRs or even an artificial 24 tyrosine polypeptide. Finally, we show that SS18 mediates BAF assembly through phase separation to regulate PST. These studies suggest that SS18 plays a role in the pluripotent to somatic interface and undergoes liquid-liquid phase separation through a unique tyrosine-based mechanism.

[1] CAS Key Laboratory of Regenerative Biology, South China Institute for Stem Cell Biology and Regenerative Medicine, Guangzhou Institutes of Biomedicine and Health, Chinese Academy of Sciences, Guangzhou, China. [2] Guangdong Provincial Key Laboratory of Stem Cell and Regenerative Medicine, Guangzhou Institutes of Biomedicine and Health, Chinese Academy of Sciences, Guangzhou, China. [3] University of Chinese Academy of Sciences, Beijing, China. [4] School of Physical Science and Technology, Lanzhou University, Lanzhou, China. [5] GMU-GIBH Joint School of Life Sciences, Guangzhou Medical University, Guangzhou, China. [6] School of Life Sciences, University of Science and Technology of China, Hefei, China. [7] Bioland Laboratory (Guangzhou Regenerative Medicine and Health Guangdong Laboratory), Guangzhou, China. [8] Laboratory of Cell Fate Control, School of Life Sciences, Westlake University, Hangzhou, China. [9] These authors contributed equally: Junqi Kuang, Ziwei Zhai. ✉email: liu_jing@gibh.ac.cn; peiduanqing@westlake.edu.cn

At the start of mammalian development, a fertilized egg is totipotent, then undergoes successive cell divisions to become a blastocyst from which naive pluripotent stem cells can be derived[1–5]. Then, blastocysts undergo implantation to mark the beginning of fetal development for both the fetus and the placenta. During or immediately after implantation, cells from the inner cell mass in a blastocyst begin to differentiate and eventually endow the three germ layers and germ cells. While this process occurs without any appreciable difficulty in vivo and naive ESCs undergo spontaneous differentiation once cultured under conditions without LIF or other supporting factors[2,6–8], one may assume that little regulatory mechanism is required for or built into this process.

Given the precision nature during mammalian development, one would argue that each cell division is a fate decision process meticulously regulated. One such example is the pluripotent to somatic transition or PST during early embryonic development. Derived from inner cell mass or ICM of mouse blastocysts[1,5], mESCs are in vitro copies of ICM, representing the pluripotent state quite faithfully by virtue of their ability to regenerate all fetal tissues when reintroduced to blastocysts even after prolong culture and storage in vitro[9]. In vitro, mESCs are capable of differentiating into cells for all lineages as well[3,10]. *Tcf3* (a.k.a. *Tcf7l1*) has been reported to play a dominant role in pluripotency exit by mediating lineage commitment[11–14]. On the other hand, *Tfe3*, a master regulator of lysosomal biogenesis[15], mediates pluripotency exit by targeting *Esrrb* through stage-specific subcellular localization[16]. In addition, pluripotency exit can be regulated at various levels, including but not limited to transcription factors[11,12,16], signal transduction pathway[12,16,17], epigenetics[18], lincRNAs[19]. While these sophisticated methods and elegant models used for mESC differentiation in vitro and in vivo, mechanistic insights at the molecular and chromatin level are still lacking at the present.

The ATP-dependent chromatin remodeling complexes BAFs (Brg/Brahma-associated factors), are also mammalian switch/sucrose nonfermentable (SWI-SNF) initially found in yeast, that consist of 15 subunits encoded by 29 genes, have been shown to play critical roles in self-renewal and pluripotency of ESCs, development[20] and human tumor[21]. Among these subunits, BRG1 is the enzyme core of BAF complexes and has been shown to be required for the maintenance of stem cells[22–24] and DPF2, as the canonical BAF specific subunit, is required for maintaining pluripotency and ESCs differentiation[25]. However, it remains to be determined if any of the BAFs are required for pluripotency exit.

CRISPR-Cas9 is an emerging technology that has been adopted for whole genome screen[26]. In this report, we describe a PST system based on inducible over-expression of cJUN in naive mESCs and its adaption for whole genome screening to identify regulators for PST. We identify SS18, a component of BAFs, as a regulator at the pluripotent-somatic interface.

## Results

**Genome-wide Cas9 screen identifies factors regulating PST.** Mouse ESCs or mESCs are naïve pluripotent stem cells capable of self-renewal indefinitely while preserving the capacity to differentiate into any of the three somatic germ layer lineages, thus representing an ideal system to study pluripotent to somatic transition or PST. To identify regulators for PST, we took advantage of our early finding that cJUN behaves as a guardian for the somatic fate in MEFs and is capable of driving mESCs to differentiate rapidly[27] (Supplementary Fig. 1a–c). Using this inducible system, we further show that exposure to cJUN for up to 6 h results in about 50% of the starting mESCs capable of regaining naive pluripotency (Supplementary Fig. 1d). However,

exposure beyond 8 h leads to <15% of the cells regaining the naive state, suggesting that there is a window of sensitivity between 6–8 h that marks the point of no return or a checkpoint between pluripotent and somatic states (Supplementary Fig. 1d, e). RNAseq analysis further reveals rapid down- and up- regulation of pluripotent genes and somatic ones between 4–8 h, respectively (Supplementary Fig. 1f, g), thus confirming the observed fate transition at the molecular level.

We hypothesize that cJUN accomplishes this rapid PST by cooperating with a cellular machinery critical to cell fate transitions. To gain insight into such machinery, we design a CRISPR based screening system as illustrated (Fig. 1a, Supplementary Fig. 1h). The rationale behind this design is our prediction that any colonies become resistant to cJUN induced PST must harbor a guide RNA targeting a critical component of this machinery (Fig. 1a). We first performed quality control tests to assure proper execution of the screens (Fig. 1b, c, Supplementary Fig. 1i). We then performed the screen as illustrated and identified multiple hits (Fig. 1d). Among the hits, SS18 scored high (Fig. 1d) and can be validated by knockout experiments with the corresponding sgRNA sequences (Fig. 1e) as measured by Nanog expression during cJUN induced PST. Interestingly, *p53* appears to play a role in PST, in addition to *Ss18* (Fig. 1e). We then performed a second screen and identified similar hits plus Brg1 (also known as Smarca4) (Supplementary Fig. 1j). Since both SS18 and BRG1 are subunits of the BAF complex[28], we decided to focus on SS18 for further analysis.

**SS18 regulates cJUN-induced PST.** Unlike *p53*[29], *Ss18* has not been reported to be involved in pluripotency nor mESC differentiation. To probe its likely function, we generated *Ss18*⁻/⁻ mESCs as illustrated in Fig. 2a. We obtained KO cell lines (Fig. 2b) that can be confirmed by western blotting for SS18 protein (Supplementary Fig. 2a). *Ss18*⁻/⁻ mESCs are morphologically identical to their wild type counterparts (Fig. 2b) and express similar naive genes under naive conditions (Fig. 2c, left panel). When induced to undergo PST, *Ss18*⁻/⁻ mESCs are resistant to cJUN induced PST with substantial cells maintaining *Nanog* expression at 8 h (Fig. 2c, right panel; Supplementary Fig. 2b). Over-expression of SS18 in *Ss18*⁻/⁻ cells can rescue the defect as measured by *Nanog* expression (Fig. 2d, Supplementary Fig. 2c) or morphology/GFP expression (Fig. 2g). At 6 h post induction, ~90% *Ss18*⁻/⁻ colonies remain pluripotent morphologically, while only 30% wild type ESCs can do so (Fig. 2e–g, Supplementary Fig. 2d). Taken together, these results demonstrate that SS18 deficiency confers resistance to cJUN induced PST. We further validated the role of SS18 in other differentiation systems (Supplementary Fig. 2e), such as spontaneous differentiation, naïve-primed differentiation and neuroectodermal differentiation, and show that the deficiency of SS18 delays all these differentiation processes as measured by *Nanog*/*Esrrb*/*Klf4* expression (Supplementary Fig. 2f) or morphology (Supplementary Fig. 2g).

**SS18 forms condensates in nuclei.** To probe the potential mechanisms associated with SS18 mediated PST, we carried out immunostaining experiments and show that endogenous SS18 forms punctate condensates during PST (Supplementary Fig. 3a). To help visualize SS18, we tagged it with EGFP and show that SS18-EGFP forms condensates readily when expressed in mESCs. We then analyzed the response of the condensates to 3% 1,6-hexanediol, an aliphatic alcohol which could disrupt weak hydrophobic interactions specifically[30] and show that it can effectively disperse SS18-GFP condensates in 10 s, and more than half of the condensates recover after 1,6-hexanediol removal in 10 s (Supplementary Fig. 3b, c). To characterize the condensates in

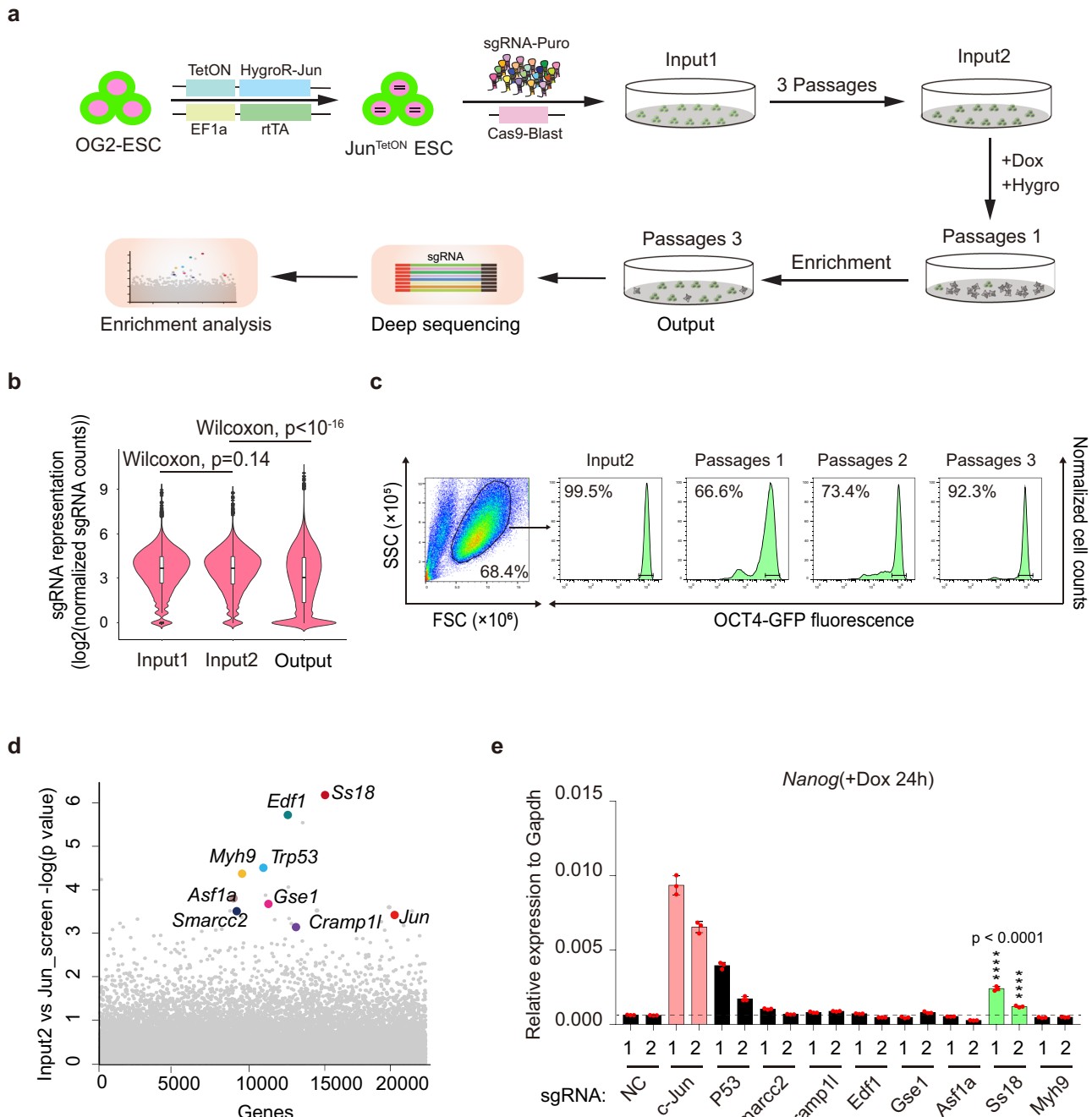

**Fig. 1 Genome-wide screen for factors regulating PST. a** The workflow for GeCKO screening strategy by HygroR-Jun[TetON] OG2ES. TetON cell line was successively infected with viruses expressing Cas9 and sgRNA. Dox and hygromycin were added after three cell passages. Dox and hygromycin were removed in 24 h and 48 h, respectively. Cells that survived after three passages were collected and analyzed. Hygro, hygromycin; HygroR, hygromycin resistance gene; GeCKO: Genome-wide CRISPR-Cas9 knockout. **b** Violin plot showing the distribution of sgRNA frequencies at different stages of screen. The distribution of sgRNA library was significantly polarized after the screening (output) compared to input 1 and input 2. Two-sided Wilcoxon test adjusted for multiple comparisons. $n = 65959$ sgRNAs. Input 1, Minimum 0, 25%Percentile 2.672, Median 3.693, 75%Percentile 4.452, Maximum 8.934; Input 2, Minimum 0, 25%Percentile 2.611, Median 3.672, 75%Percentile 4.456, Maximum 8.764; Output, Minimum 0, 25%Percentile 1.366, Median 3.064, 75%Percentile 4.391, Maximum 10.07. Two independent experiments. **c** Distribution of OG2-GFP signaling at different passages by FACS during this screening. **d** Identification of top candidate genes using the MAGeCK[43] algorithm. **e** Validation of the hits in (**d**) by corresponding sgRNAs within GeCKO library during c-JUN induced PST. Data are mean ± s.d., two-sided, unpaired *t* test; $n = 3$ independent experiments, ****$p < 0.0001$.

more detail, we performed FRAP experiments, and show that SS18-EGFP condensates recover rapidly in ~10 s after bleaching (Fig. 3a, Supplementary Fig. 3d). We further bleached the SS18-GFP condensates in the edge (Case1) or middle (Case 2) and observed a quick florescence recovery within 6 s in both conditions (Fig. 3b). When we measured the pixel intensity along the white line in case 2, we can show a clear recovery in kymography (Fig. 3c). We then quantified the mobility of SS18 protein by measuring the pixel intensity during fluorescence recovery inside or outside of eight bleached regions within condensates, and show that $\tau$(inside) = 1.841 s, $\tau$(outside) = 1.905 s; Plateau(inside) = 66.18%, Plateau(outside) = 79.91% (Fig. 3d), suggesting a quick movement

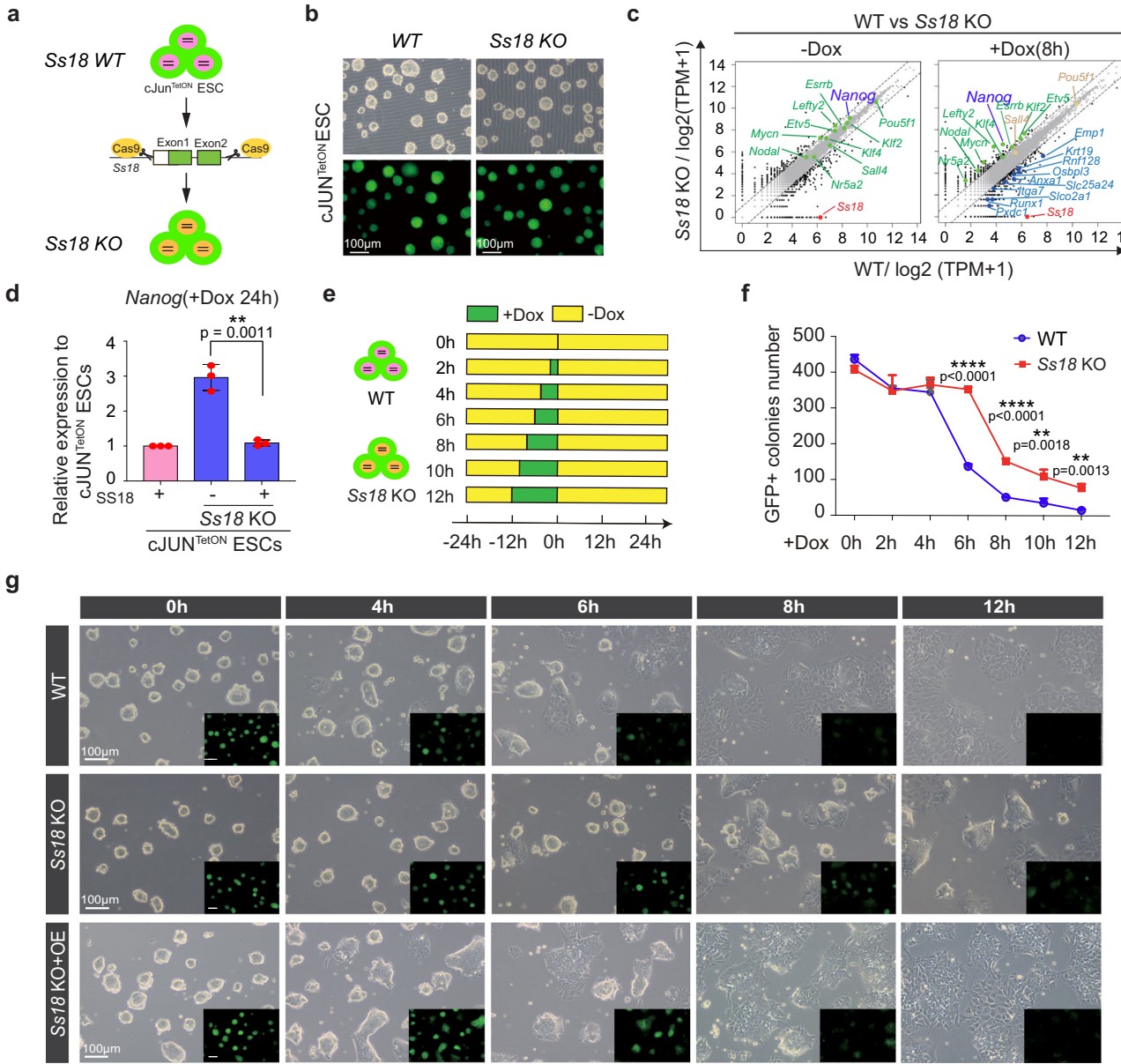

**Fig. 2 SS18 Regulates cJUN-induced PST. a** Construct design for SS18 knockout in mESC. The cleavage sites of Cas9 were designed flanking exon1 and exon2 within *Ss18* loci. **b** SS18KO mESCs are similar to WT ones in morphology. Three biological replicates. **c** Gene expression profiles by RNA-seq for WT and SS18 KO mESCs. SS18 knockout has little impact on most pluripotent genes but delays the c-JUN induced PST process significantly. **d** Rescue of the delayed PST as indicated by Nanog expression in SS18 KO mESCs with SS18 in lentiviral vector. Data are mean ± s.d., two-sided, unpaired *t* test; *n* = 3 independent experiments, **$p < 0.01$. **e** The experimental design to test the window of sensitivity for SS18 response to c-Jun-induced PST. **f** Results from (**e**) to show that OCT4-GFP positive colony numbers recovered for 48 h in 2i/LIF medium after c-Jun induction at the indicated time points for wild type and SS18 KO ESCs. Data are mean ± s.d., two-sided, unpaired *t* test; *n* = 3 independent experiments. **$p < 0.01$, ****$p < 0.0001$. Note no change for 4 h and the greatest change at 6 h for the window of sensitivity. **g** Representative images for the c-Jun-induced PST in WT, *Ss18*$^{-/-}$ and *Ss18*$^{-/-}$ overexpressing *Ss18*, respectively. Three biological replicates.

of SS18 in the condensates. Moreover, the condensates appear to be able to divide or fuse quite rapidly as well (Fig. 3e). We further performed quantitative analysis of fusion and fission events for the condensates in cells (Supplementary Fig. 3e) and show that the time scale for SS18-EGFP condensate fusion increases almost linearly with their diameters, resulting a fitting slope K = 0.65 ± 0.19. These data suggest that SS18 forms liquid-like condensates readily[31].

**SS18 has IDRs critical for its activity.** Proteins that undergo liquid-liquid phase separation to form punctate condensates as we described for SS18 often contain IDR or Intrinsically Disordered Region[30,32–35]. To test if SS18 has similar IDR, we analyzed amino acid sequence of SS18, and show that except the first 70 residues at the N terminus, the rest 71-418 residue domain have a strong IDR (Fig. 4a). We then constructed a series of mutants for the IDR (Supplementary Fig. 4a), confirming their expression by western blot (Supplementary Fig. 4b), and show that its ability to rescue *Ss18*$^{-/-}$ mESCs is dependent on the length of IDR left in the mutants (Supplementary Fig. 4c). Based on the rescue activities, we reason that IDR223-379 may be most critical (pink box in Fig. 4b). This sub region appears to be enriched with Q, Y, P and G (Fig. 4b). We

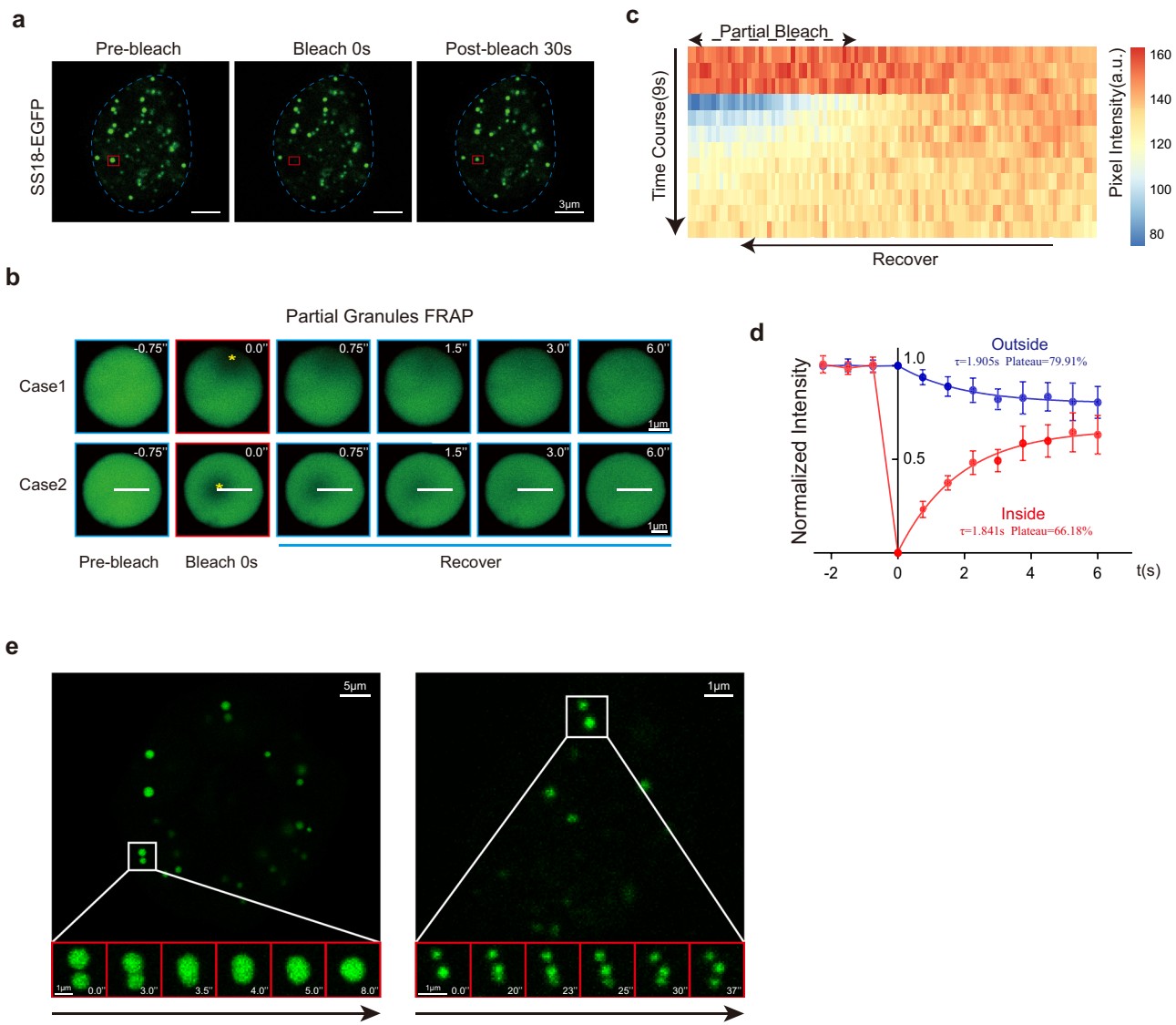

**Fig. 3 SS18 forms condensates. a** Representative images of FRAP experiments with lentiviral SS18-EGFP expressed in mESCs. The droplet undergone bleaching was indicated in the red box. The blue dotted line represents the nuclear boundary. Scale bars,3 μm. laser intensity 2.5%, detector gain 500 V, background subtraction treatment. Seven condensates from two independent experiments. **b** Time course microscopy images of fluorescence recovery in SS18-EGFP condensates after partial photobleaching at the edge (case 1) and middle (case 2). SS18-EGFP was expressed by transient transfection in 293 cells to obtain the large enough condensates. The bleached regions were marked by yellow asterisks. Scale bars, 1 μm. laser intensity 2.5%, detector gain 500 V, background subtraction treatment. Eight condensates from two independent experiments. **c** Kymography for the time course of pixel intensity along the white line in the condensate of case 2 in (**b**). The intensity recovered in the bleached region inside the condensate as the unbleached part decreased. Pixel number = 103. **d** The intensity dynamics inside and outside of the bleached regions within condensates ($n = 8$) was fit to an exponential function. t = 0 was the bleached time point. $\tau$(inside) = 1.841 s, $\tau$(outside) = 1.905 s; Plateau(inside) = 66.18%, Plateau(outside) = 79.91%, diffusion coefficient on the order of D ~ $L^2/\tau$ ~5 μm$^2$/s, L is the diameter of the bleached region. Data are mean ± s.d., two-sided, unpaired $t$ test; $n = 8$ condensates. **e** Representative droplet fusion events are shown by time course images. SS18-EGFP was expressed by transient transfection to obtain the large condensates as in (**b**). Scale bars, 2 μm. laser intensity 2.5%, detector gain 500 V, background subtraction treatment. Nine fusion/fission cases.

then constructed four mutants, Q-A, Y-A, P-A and G-A by substituting each of those residues with A (Fig. 4b, Supplementary Fig. 4d). Among them, Y-A mutant lost its ability to form condensates, while the other three mutants appear to be retaining the ability to form condensates, compared to the WT SS18 or the truncated form SS18-IDR (Fig. 4c, d, Supplementary Fig. 4e–g). Importantly, we show that this Y-A mutant fails to rescue $Ss18^{-/-}$ mESCs in PST (Fig. 4c). We further investigated the co-localization of SS18 or mutants with BRG1 by immunostaining and pixel intensity analysis, and show that except Y-A mutant, all the other three mutants can colocalize with BRG1

in the same condensates (Fig. 4d, e). Surprisingly, we find that SS18-IDR (71-418) is sufficient to form condensates, but unable to colocalize with BRG1 (Fig. 4d, e), suggesting that the N-terminal residues are not required for condensates formation, but critical for recruiting BRG1. We further performed IP-MS experiments to investigate whether the SS18 Y-A mutant can pull down the components in BAF complex (Supplementary Fig. 4h) and show that Y-A mutant lost the capability to interact with BAF subunits (Fig. 4f). Taking together, these results suggest that SS18 contains a Y-type IDR required for BRG1 interaction and PST.

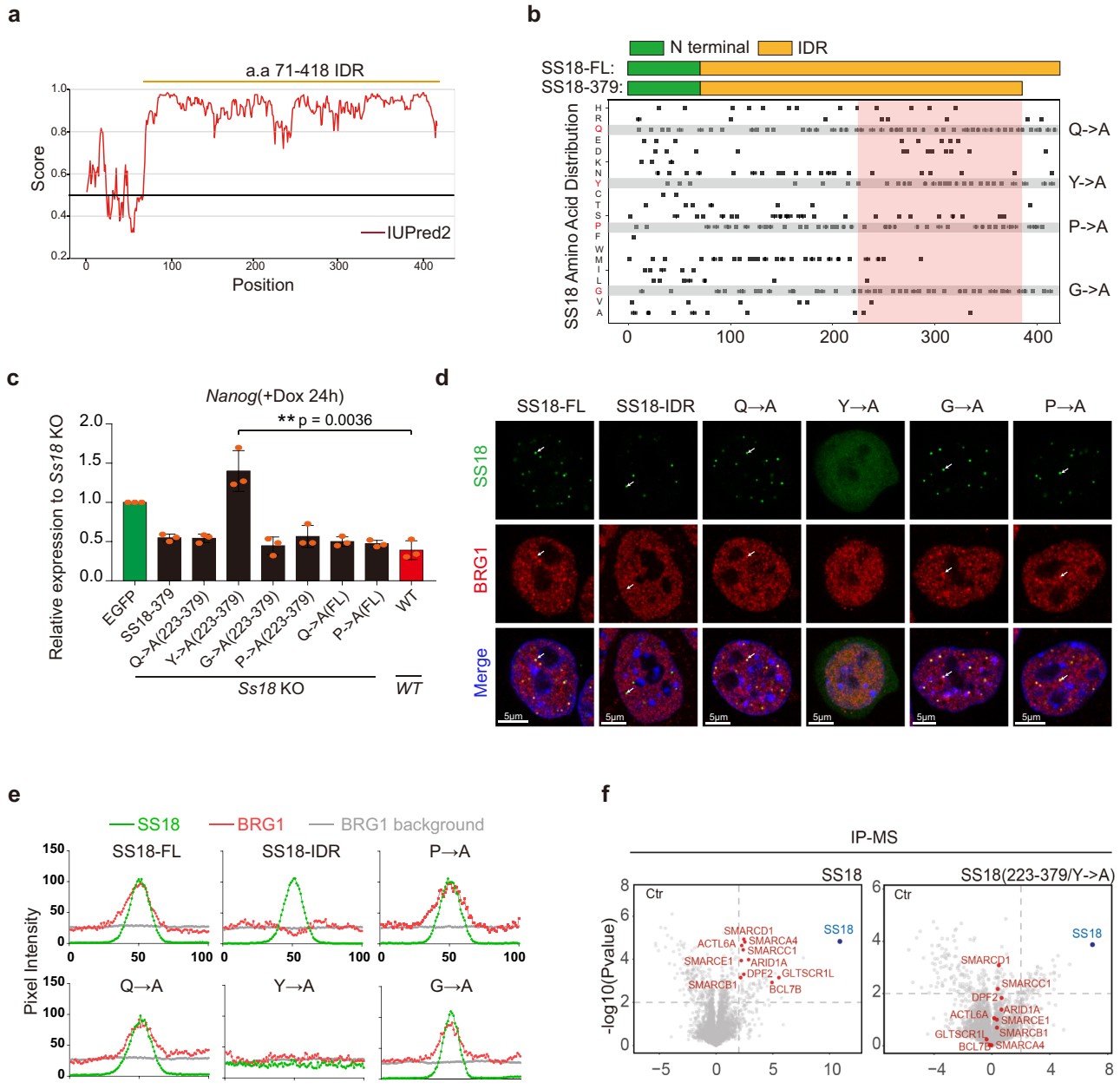

**Fig. 4 SS18 condensates through a tyrosine-based mechanism. a** Graph plotting intrinsic disorder of SS18 by the IUPred2A algorithm (https://iupred2a. elte.hu/). IUPred2 scores are shown on the y axis, and the amino acid positions are shown on the x axis. **b** Schematic illustration for Ss18 truncations shown in the upper panel with the amino acid distribution of SS18 indicated in the lower panel. The pink box highlights the necessary region of C-terminal for the function of SS18 in PST. Q->A, Y->A, P->A, G->A indicate the amino acid Glutamine, Tyrosine, Proline, Glycine mutated to Alanine, respectively. **c** The expression of *Nanog* in *Ss18*[−/−] mESCs overexpressing the indicated Ss18 mutants during c-Jun based PST. Note the failure of Y-A to rescue PST. Data are mean ± s.d., two-sided, unpaired *t* test; *n* = 3 independent experiments, **\*\****p* < 0.01. **d** Representative images for the indicated WT, IDR and SS18-mutant-EGFP fusion proteins and immunofluorescence imaging for BRG1 in mESCs. Scale bars, 5 μm. Note the failure of Y-A mutant to form condensates. SS18-EGFP, laser intensity 2.5%, detector gain 500 V. BRG1 immunofluorescence, laser intensity 3.5%, detector gain 750 V. Both SS18-EGFP and BRG1 immunofluorescence were processed by background subtraction. Eight condensates. **e** The graphs showing the pixel intensity in Fig. 4d. The green curves indicate the pixel intensity dynamic across six condensates of various SS18 mutants and the corresponding pixel intensity of BRG1 indicated by the red curves; the average pixel intensity of BRG1 within nuclei and outside of the nucleoli were indicated by gray curves. **f** Volcano plots of IP-MS results for FLAG tagged SS18 and SS18Y-A mutant expressed in SS18[−/−] mESCs by anti-Flag antibody with label-free quantification. SS18[−/−] cells serves as control. Every point represents a single protein. SS18 and BAF components are marked blue and red. IP-MS experiments were performed in triplicates and a two-sample *t* test was applied. *P* = 0.01 and fold change = 2 are used as threshold.

**Functional rescue of SS18 IDR.** Condensate formation through phase separation is a physic property mediated by several distinct mechanisms[34,35]. Our results from the Y-A mutant further suggest that SS18 may rely on Y residues to form condensates (Fig. 4d). To test if SS18 IDR can be replaced by other types of

IDRs, we constructed a set of fusions between the N-terminal domain of SS18 and poly-Y, S centered/enriched IDRs from MED1 and P/Q enriched IDRs from BRD4[32], Y centered/enriched IDRs from TAF15 and FUS[33] (Supplementary Fig. 5a). We show that poly-Y, TAF15-IDR and FUS-IDR, but not MED1-IDR

nor BRD4-IDR, can rescue condensates formation (Fig. 5a). Consistently, poly-Y, TAF15- and FUS- PLDs can functionally replace SS18-IDR in rescuing SS18$^{-/-}$ defect during cJUN-based PST (Fig. 5b). We further performed IP-MS experiments to investigate whether these SS18 truncation/fusions can pull down the subunits in BAF complex (Supplementary Fig. 5b), and show by pairwise comparison scatterplots that SS18 N-terminal, N-polyY and N-FUS(IDR) can pull down the subunits of BAF complex while SS18(IDR) could not (Fig. 5c). Notably, we show that N-BRD4 could pull down the subunits in BAF complex, but still fails to mediate PST (Fig. 5b, c), perhaps due to the type of condensates formed (Fig. 5a). We summarize the results into a table in Fig. 5d and show that SS18 mediates PST by forming BAF complex through its N-70 domain and condensates with a C-terminal IDR (Fig. 5e). These results further suggest that IDRs are interchangeable among the Y-type ones and can be designed artificially like the poly-Y.

**SS18 promotes BAF/ncBAF but impedes PBAF assembly during PST.** mSWI/SNF ATP-dependent chromatin remodeling complexes have three subtypes that share many core subunits such as BRG1, SMARCD1, ACTL6A and BCL7A/B/C, but differ in specific ones, i.e., the canonical BAF with DPF1, DPF2,

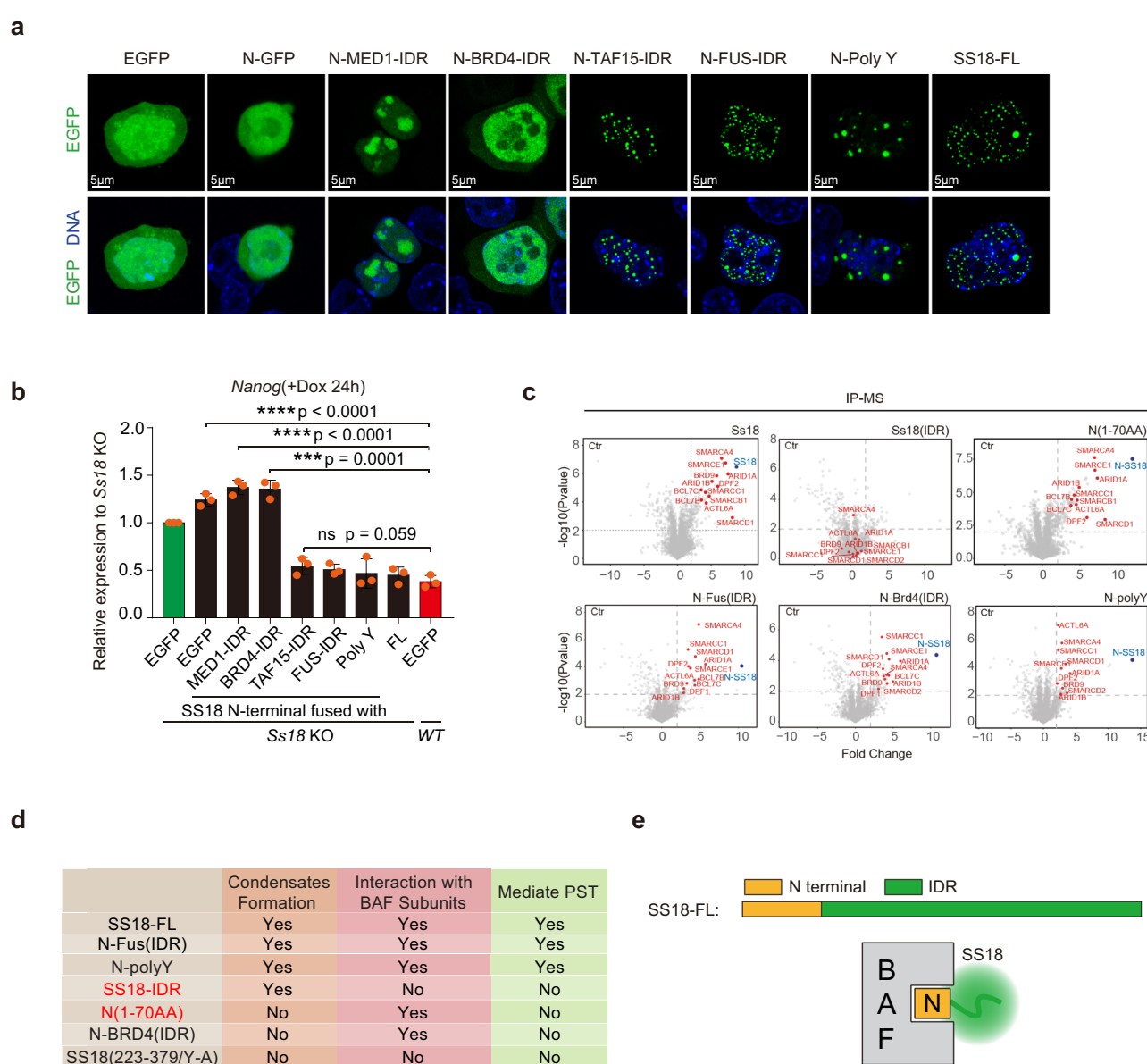

**Fig. 5 SS18 interacts with BAF through its N-terminal 70 residues. a** Representative images for EGFP fusions expressed in mESCs as indicated. Note the similar condensates formed by SS18, FUS and TAF IDRs as compared to those for MED1 and BRD4. Poly Y is an artificial 24 tyrosine residue polypeptide. FL full length. Scale bars, 5 μm. laser intensity 2.5%, detector gain 500 V, background subtraction treatment. Three biological replicates. **b** Rescue of Ss18KO by constructs in (**a**) to show that IDRs from TAF and FUS or even polyY can substitute that of SS18. Data are mean ± s.d., two-sided, unpaired $t$ test; $n = 3$ independent experiments, ***$p < 0.001$, ****$p < 0.0001$. **c** Volcano plots of IP-MS results from flag tagged Ss18-FL, SS18(IDR), N terminal 70aa of Ss18 alone or fused with IDRs from FUS or Brd4 and polyY expressed in Ss18KO mESCs by Flag antibody with label-free quantification. Every point represents a single protein. SS18 and BAF components are marked in blue and red, respectively. IP-MS experiments were performed in triplicates and a two-sample $t$ test was applied. $P = 0.01$ and fold change $= 2$ were used as threshold. **d** Summary table for various constructs for their abilities to condensate, interact with BAF and mediate PST. **e** A proposed structure and function relationship for SS18: N-terminal BAF interacting domain or BID and C-terminal condensate forming IDR domain (CFD).

ARID1A, and ARID1B, the PBAF (Polybromo-associated BAF) with PBRM1, PHF10, ARID2 and BRD7 and the recently reported ncBAF (non-canonical BAF) containing BRD9, GLTSCR1/1 L. Among them, SS18 is involved in the assembly of BAF and ncBAF (Supplementary Fig. 6a)[20,36,37].

To investigate condensates associated with SS18, we stained SS18KO mESCs expressing SS18-GFP with antibodies against endogenous BAF specific subunit DPF2 or PBAF specific subunit BRD7 and show that DPF2, not BRD7, is enriched with SS18 (Fig. 6a, b, Supplementary Fig. 6b), which is also consistent with endogenous SS18 (Supplementary Fig. 6c, d, e). These results demonstrate that SS18 is associated exclusively with BAF, not PBAF, in mESCs.

To test the role of SS18 in BAF assembly, we immunoprecipitated the common subunit BRG1 in SS18 WT and SS18 KO mESCs and then analyzed the complexes by IP-MS to detect the relative abundance of each components. We found SS18 deficiency decreases the assembly of canonical BAF and ncBAF, but promotes that of PBAF (Fig. 6c). In addition, the full length not N terminal of SS18 can rescue BAF assembly in SS18 KO mESCs (Supplementary Fig. 7a, b). Given the two unbiased screens we performed that identified both SS18 and Brg1 but no PBAF specific subunits, we speculate that PST relies on primarily BAF or/and ncBAF, not PBAF. We indeed can show that BAF ncBAF and PBAF serve different roles in PST by shRNA knockdown of BAF/ncBAF/PBAF specific subunits and shared ones (Fig. 6d, Supplementary Fig. 7c, Supplementary Data 2). The depletion of BAF specific components, DPF1/2, and ncBAF specific ones can delay this transition, while depleting PBAF specific subunit BRD7 promotes PST (Fig. 6d). Unlike Ss18, Brg1 knockdown in mESCs leads to defects in self-renewal and survival morphologically (Supplementary Fig. 7d), consistent with a previous report[22–24]. We further performed RNA-Seq experiments to investigate the downstream targets of Brg1 in mESC, and show by GO analysis that the 572 genes affected by Brg1 knockdown are involved in cell-substrate adhesion, positive regulation of cell migration and epithelial cell development (Supplementary Fig. 7e). However, BRD9 inhibitor BI-9564 has no effect in the process (Supplementary Fig. 7f). We knocked down the specific components of ncBAF and PBAF in SS18 KO mESCs and found the deficiency of ncBAF will delay PST further, on the contrary, the knockdown of PBAF will rescue that (Supplementary Fig. 7g), which suggests that BAF and ncBAF play similar/redundant roles against PBAF in cJUN induced PST.

To further investigate the mechanism underlying the different effects between BAF and PBAF within PST, we performed ChIP-seq of H3K27ac, cJUN, SS18, and PBAF specific subunits PBRM1 and BRD7 at different time points during PST (Supplementary Fig. 7h). All the H3K27ac occupancy sites ± 5 kb of 0 h and 8 h are divided into three categories (8 h down, permanent and 8 h up) according to the dynamic changes of intensity by fold change >2. Both BAF and PBAF complexes relocate to the same sites occupied by cJUN. We can estimate the relocation rate from the ratio of average intensity of up sites and down sites from the graph. Contrary to PBAF subunits, SS18 has higher relocation rate at 4 h and 8 h during PST (Supplementary Fig. 7h), suggesting that BAF containing SS18, compared with PBAF, can facilitate cell fate transition more efficiently. Taken together, these data suggest that SS18 promotes BAF but inhibits PBAF assembly to mediate PST (Fig. 6e).

## Discussion

Broadly speaking, the mammalian cell fate continuum can be divided into two categories: pluripotent and somatic. Pluripotent cells have the capacity to give rise to all somatic cells[9]. On the other hand, somatic cells can be reprogrammed to a pluripotent state[38–40]. The inter-conversion between somatic and pluripotent cells through differentiation and reprogramming suggests that there may be an interface between somatic and pluripotent states. We have proposed previously that cJUN serves as a guardian of somatic fate as opposed to factors such as Oct4 that specify the pluripotent fate[27]. In this report, we further define a molecular link at the pluripotent-somatic interface. Mechanistically, we propose that cJUN drives a rapid transition through this interface with the help of SS18, a well-known member of the BAF complex. We provided evidence that SS18 is a bipartite protein with an N-terminal 70aa domain specific for BAF interaction and a C-terminal IDR for forming condensate through a unique tyrosine-based mechanism.

It is of interest to note that the IDR of SS18 can be replaced by similar IDRs from FUS and TAF15[33], but not BRD4 and MED1[32] (Fig. 5b). We noted that the IDRs of FUS and TAF15, like that of SS18, are enriched with tyrosine residues, while those of BRD4 and MED1 are not (Supplementary Fig. 5a). Based on these observations, we further reasoned that perhaps a polymer of tyrosine residues can also function as an IDR to form condensates. Indeed, when we replaced the IDR of SS18 with a 24-tyrosine residue polypeptide, similar condensates can be also observed (Fig. 5A, N-Poly Y). This construct can also rescue full length SS18 function in PST assays (Fig. 5b). Based on these results, we would like to propose that the 24-residue polymer can serve as a model tool to further analyze liquid-liquid phase separation or condensate formation in vitro and in vivo. The simple composition as well as well-behaved pattern of condensates may help further illuminate principles and rules associated with phase separation. More generally, it may be possible to design additional polymers that behave similarly to other classes of IDRs such as those from BRD4 or MED1.

Chromosome translocation t(X;18) (p11.2; q11.2) has been suggested as a cause of synovial sarcoma[28]. Our dissection of its structure into two main domains, i.e., the N-terminal BAF interacting domain and the C-terminal IDR, may further encourage innovative strategies to design targeted therapies.

## Methods

**Cell culture**. HEK293T cells are cultured in DMEM supplemented with 10% FBS, GlutaMAX and NEAA. Mouse ESCs are maintained under feeder-free condition with mESC-2i/LIF medium containing DMEM, 15% FBS, NEAA, GlutaMAX, PD0325901 (Targetmol T6189), Chir99021 (Targetmol T2310), LIF (Millipore ESGE107). HEK293T was obtained from ATCC (CRL-1126). mESCs were derived in-house by crossing Oct-GFP trans genetic allele carrying male mice (male CBA/cAJ x female C57bl/6 J) and female 129/Sv mice. All the cell lines have been confirmed as mycoplasma free with Lonza LT07-318. For random differentiation, mouse ESCs were dissociated and plated at 2 × 104/cm2 in mESC medium without LIF and analyzed after 24 h incubation. For Naïve-Primed differentiation, mouse ESCs were dissociated and plated at 2 × 104/cm2 in mESC-2i/LIF medium. After 24 h medium was replaced with FA medium (50% DMEM/F12 (GIBCO), 50% Neurobasal (GIBCO), 0.5× N-2 (GIBCO), 0.5× B-27 (GIBCO), 1% GlutaMAX (GIBCO), 1% non-essential amino acids (GIBCO), 1% sodium pyruvate (GIBCO), 0.1 mM 2-mercaptoethanol (GIBCO), 15 ng/ml bFGF (PeproTech), 20 ng/ml Activin A (PeproTech) and 1 μM XAV939 (Selleck S1180)) for a further 24 h prior to analysis. For neuroectodermal differentiation, mouse ESCs were dissociated and plated onto 0.2% gelatin-coated plastic at a density of 2 × 104/cm2 in N2B27 medium (50% DMEM/F12 (GIBCO), 50% Neurobasal (GIBCO), 0.5× N-2 (GIBCO), 0.5× B-27 (GIBCO), 1% GlutaMAX (GIBCO), 1% non-essential amino acids (GIBCO), 1% sodium pyruvate (GIBCO) and 0.1 mM 2-mercaptoethanol (GIBCO)) and analyzed after 24 h incubation.

**Gene knockout in mESCs**. Knockout mouse ESCs lines were generated using the CRISPR/Cas9 system[41,42]. Guide RNAs were designed using crispr.mit.edu. and cloned downstream of the human U6 promoter of pX330 (Addgene). For SS18 KO in mouse ESCs, two pairs of Guide RNAs were designed for each gene to delete critical exons. For Ss18, the gene locus of exon1 to exon2 were deleted. Mouse ESCs clones carrying deletions in both alleles were identified by genotyping and western blot.

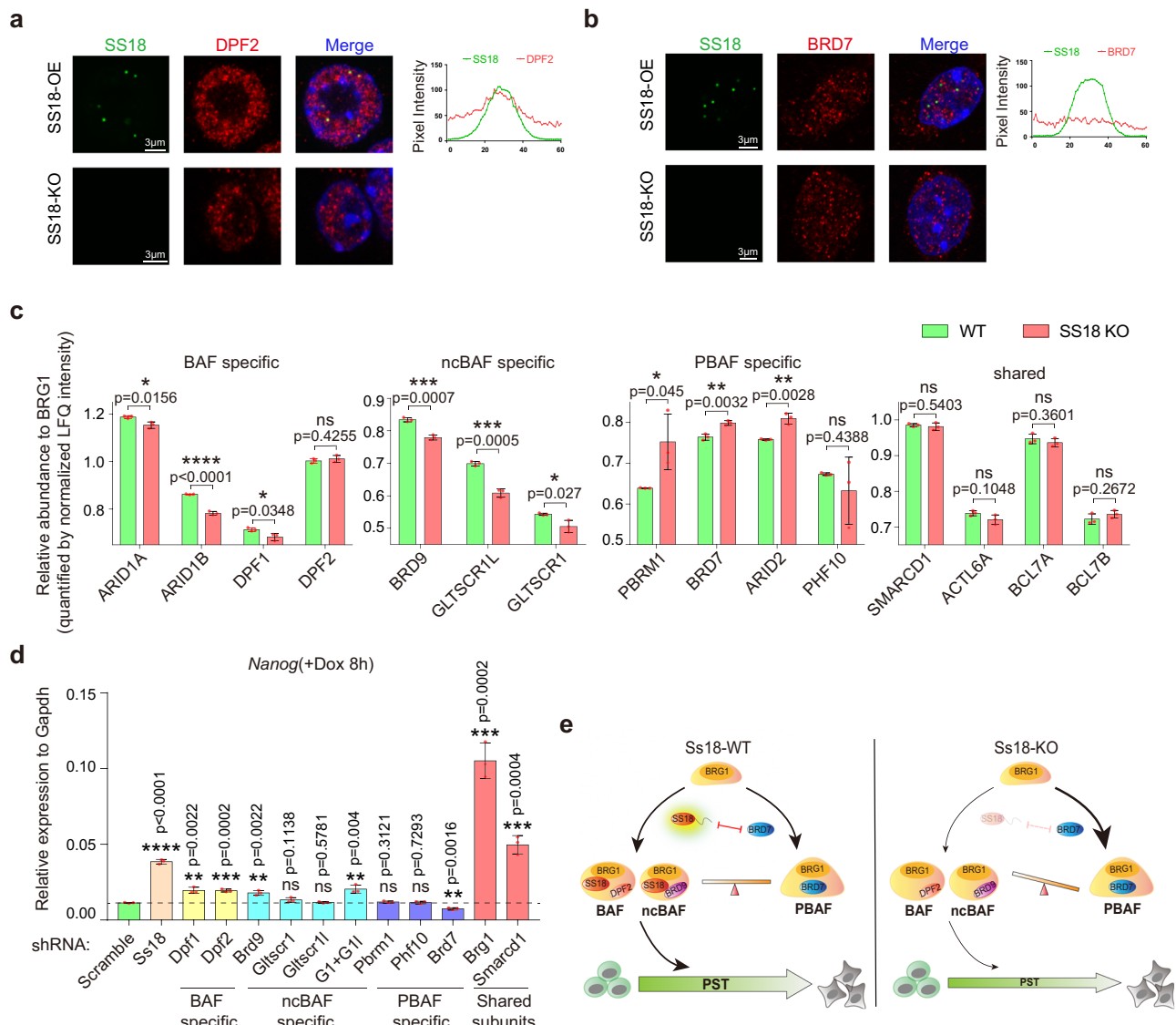

**Fig. 6 SS18 regulates PST through mediating the assembly balance among BAF ncBAF and PBAF. a** Representative images of immunofluorescence for endogenous BAF specific subunit DPF2 in SS18KO mESCs infected with lentiviral SS18-EGFP. The graph on the right shows pixel intensity across five condensates of SS18-EGFP (green) and DPF2 (red). Scale bars, 3 μm. SS18-EGFP, laser intensity 2.5%, detector gain 500 V. DPF2 immunofluorescence, laser intensity 3.5%, detector gain 750 V. Both SS18-EGFP and DPF2 immunofluorescence were processed by background subtraction. Three independent experiments. **b** Representative images of the immunofluorescence for endogenous PBAF specific subunit BRD7 in SS18KO mESCs with exogenous SS18-EGFP. The graph on the right shows pixel intensity across five condensates of SS18-EGFP and BRD7 (red). Note no co-localization between SS18 and BRD7. Scale bars, 3 μm. SS18-EGFP, laser intensity 2.5%, detector gain 500 V. BRD7 immunofluorescence, laser intensity 3.5%, detector gain 750 V. Both SS18-EGFP and BRD7 immunofluorescence were processed by background subtraction. Three independent experiments. **c** The histogram shows the relative precipitated abundance of indicated proteins to BRG1 as quantified by normalized LFQ intensity from IP-MS experiments by anti-BRG1 antibody performed on WT and SS18KO mESCs. IP-MS experiments were performed in triplicates and a two-sample *t* test was applied. Data are mean ± s.d., two-sided, unpaired *t* test; *n* = 3 independent experiments, *$p < 0.05$, **$p < 0.01$, ***$p < 0.001$, ****$p < 0.0001$. **d** The expression levels of Nanog by Jun-FlagTetON mESCs at 8 h during PST with knocking down of indicated components of BAF ncBAF or PBAF. G1 + G1l, knockdown Gltscr1 and Gltscr1l jointly. Data are mean ± s.d., two-sided, unpaired *t* test; *n* = 3 independent experiments, **$p < 0.01$, ***$p < 0.001$, ****$p < 0.0001$. **e** A working model for SS18-mediated PST. SS18 functions by interacting with BAF or ncBAF subunits through its N-terminal 70aa, and forming condensates through a tyrosine-rich IDR. Meanwhile, the full length of SS18, co-localizing with BAF-specific component and isolating PBAF-specific one, can promote the BAF complex assembly and impair that of PBAF. In WT cells, BAF ncBAF and PBAF reach a balance. In SS18KO cells, the balance shifts towards PBAF to cause a delay in PST.

**Immunoblotting**. Cells were collected and lysed in lysis buffer supplemented with protease inhibitor cocktail (Roche) on ice for 15 min, and then cells were boiled in 100 °C for 10 min. After centrifugation, the cell supernatants were subjected to SDS–PAGE and incubated with corresponding primary antibody and secondary antibodies. The following antibodies were used in the project: anti-SS18 (CST no. 21792. 1:1000), anti-EGFP (CST no. 2555 s, 1:1000), anti-GAPDH (KangChen Biotech, no. KC-5G5, 1:1000), anti-JUN (abcam, no. ab31419, 1:1000), anti-BAF155 (Santa Cruz Biotechnology, no. sc-32763, 1:500).

**Immunofluorescence**. Cells growing on coverslips were washed three times with PBS, then fixed with 4% PFA for 30 min, and subsequently penetrated and blocked with 0.1% Triton X-100 and 3% BSA for 30 min at room temperature. Then, the cells were incubated with primary antibody for half hour. After five washes in PBS, 1 h of incubation in secondary antibodies, cells were then incubated in DAPI (Sigma-Aldrich D9542) for 2 min. Then, the coverslips were mounted on the slides for observation on the confocal microscope (Zeiss LSM 900 with Airyscan detector). The following antibodies were used in this project: anti-SS18, (CST no. 21792.

1:1000). anti-BRG1, (CST no. 52251 1:1000). anti-DPF2, (Proteintech, no. 12111-1-AP-50 µl,1:100). anti-BRD7, (Proteintech, no.51009-2-AP-50 µl, 1:100).

**qRT–PCR and RNA-seq.** Total RNAs were prepared with TRIzol. For quantitative PCR, cDNAs were synthesized with ReverTra Ace (Toyobo) and oligo-dT (Takara), and then analysed by qPCR with ChamQ SYBR qPCR Master Mix (Vazyme). The qRT-PCR primers used in this study are provided in Supplementary Data 1. VAHTS mRNA-seq V3 Library Prep Kit for Illumina (NR611, Vazyme) was used for library constructions and sequencing done with NextSeq500 Mid output 150 cycles (FC-404-2001, Illumina) for RNA-seq. RNA-seq data was analyzed by DEseq2, GO.db and Mfuzz.

**Genome-wide CRISPR-Cas9 screens.** The lentiviral gRNA plasmid library and CRISPR/Cas9 expressing plasmid for genome-wide CRISPR-Cas9 screen were obtained from Addgene (#1000000053). Amplification of the library and preparation of the lentivirus were performed following the protocol provided by Addgene with electrocompetent E. coli (TaKaRa). HygroR-cJunTetONESCs stably expressing Cas9 were seeded in the 15-cm dishes (about $3 \times 10^6$ cells per dish) and a total of $2.5 \times 10^7$ cells were infected with the gRNA lentivirus library to keep the diversity of library. Meanwhile, the MOI of infection was <0.3. Puromycin (1 µg/ml GIBCO A1113802) was used to eliminate non-infected cells after 48 h. The infected cells were collected as the sample of input 1 and the cells of further three passages were collected as the sample of input 2. Next, 2 µg/ml doxycycline was used to trigger the expression of HygroR-cJun fusion protein and 10 µg/ml hygromycin was added in the culture medium to eliminate the cells without response to doxycycline. Dox was added only for the first 24 h at passage1 to drive PST but withdrew in the following three passages. The GFP positive cells were analyzed by FACS when passed in 24 h after Dox withdrawal. Almost all the cells would have transitioned from pluripotency to somatic state. The cells resisting PST or apoptosis were enriched for three passages in mESC medium, while the rest ones were eliminated gradually. Finally, the enriched cells were collected as the sample of output group.

**Illumina sequencing of gRNAs and statistical analysis.** The genomic DNA of the three samples (input 1, 2 and output group) were extracted and the Illumina sequencing of gRNAs were conducted as described previously[26]. The counts of unique sgRNA for a given sample were normalized as follows:

$$\text{normalized sgRNA counts} = \frac{counts\,of\,per\,sgRNA}{the\,sum\,of\,counts\,of\,all\,sgRNA} \times 10^6 + 1 \quad (1)$$

Enrichment and depletion of guides and genes were analysed using MAGeCK statistical package5[43] by comparing read counts between input 2 and input 1 or output group and input 2.

**FACS analysis.** For OCT4-GFP analysis in the process of screen, cells were suspended in DPBS supplemented with 2% FBS (FACS buffer) for direct detection. DAPI was used to exclude the nonviable cells. The cells were then analyzed on Accuri C6 flow cytometer (BD Biosciences). The GFP fluorescence intensity was detected in the FITC channel. Data analysis was done using FlowJo7.6.1. software (LLC).

**ChIP-Seq.** Briefly, cells were fixed with 1% formaldehyde for 10 min and then followed by the reaction with 0.125 M glycine. Cells were then lysed in ChIP-buffer A (50 mM HEPES-KOH, 140 mM NaCl, 1 mM EDTA (pH 8.0), 10% glycerol, 0.5% NP-40, 0.25% Triton X-100, 50 mM Tris-HCl (pH 8.0), and protease inhibitor cocktail) for 10 min at 4 °C. Samples were centrifuged at 1400 g for 5 min at 4 °C. Pellets were lysed in ChIP-buffer B (1% SDS, 50 mM Tris-HCl (pH 8.0), 10 mM EDTA and protease inhibitor cocktail) for 5 min at 4 °C. The DNA was fragmented to 100–500 bp by sonication, and then centrifuged at 12,000 g for 2 min. The supernatant was diluted with ChIP IP buffer (0.01% SDS, 1% Triton X-100, 2 mM EDTA, 50 mM Tris-HCl (pH 8.0), 150 mM NaCl and protease inhibitor cocktail). Immunoprecipitation was performed with 1 µg anti-H3K27ac antibody (Abcam, ab4729) coupled to Dynabeads with proteinA/G overnight at 4 °C. Beads were washed, eluted and reverse crosslinked. DNA was extracted by phenol/chloroform for sequencing. The ChIP DNA library was constructed following the Illumina ChIP-seq library generation protocol. Briefly, 5 ng ChIP DNA was blunt-ended, and then a dA tail was added. Illumina genomic adapters with index sequences were ligated to the DNA. The adapter-ligated DNA was amplified by PCR for 18 cycles, and the resulting DNA libraries were quantified and tested by qPCR with positive primers to assess the quality of the library. The ChIP DNA library for NextSeq 500 sequencing was constructed with VAHTS Turbo DNA Library Prep Kit for Illumina (Vazyme) according to manufacturer's instructions. The ChIP-seq for other factors in mESCs was constructed with NovoNGS CUT&Tag 2.0 High-Sensitivity Kit for Illumina (novoprotein N259-YH01) according to manufacturer's instructions. ChIP-seq data are mapped to the 10 mm mouse genome assembly using bowtie2, version 2.4.1.

**SS18 endogenous immunoprecipitation and on-bead digestion.** Whole cell extracts of mES cells with cJun overexpression were prepared using lysis buffer (50 mM Tris pH 8.0, 150 mM NaCl, 10% Glycerol, 0.5% NP40) with fresh added 1x Complete Protease inhibitors (Sigma, 1187358001). Cells were incubated for 2 h at 4 °C on a rotation wheel. Soluble cell lysates were collected after maximum speed centrifugation at 4 °C for 15 min. 1 mg of cell lysates were incubated with either SS18 antibody or matched IgG overnight at 4 °C on a rotation wheel. Combined Protein A/G magnetic beads (Bio-rad, 1614833) were added for another 1.5 h. Beads were then washed three times with wash cell lysis buffer and one time with PBS. After completely removal of PBS, immunoprecipitated proteins were digested using on-bead digestion protocol as described before 17. Briefly, beads were incubated with 100 mL of elution buffer (2 M urea, 10 mM DTT and 100 mM Tris pH 8.5) for 20 min. Afterwards, iodoacetamide (Sigma, I1149) was added to a final concentration of 50 mM for a 10 min in the dark, following with 250 ng of trypsin (Promega, V5280) partially digestion for 2 hr. After incubation, the supernatant was collected in a separate tube. The beads were then incubated with 100 mL of elution buffer for another 5 min, and the supernatant was collected in the same tube. All these steps were performed at RT in a thermo shaker at 500 g. Combined elutes were digested with 100 ng of trypsin overnight at RT. Finally, tryptic peptides were acidified to pH < 2 by adding 10 mL of 10% TFA (Sigma, 1002641000) and desalted using C18 Stage tips (Sigma, 66883-U) prior to MS analyses. Each experiment was performed in technical triplicate.

**Mass spectrometry analysis.** Tryptic peptides were separated by AcclaimTM PepMapTM 100 C18 column (Thermo, 164941) using a 140 min of total data collection (100 min of 2–22%, 20 min 22–28% and 12 min of 28–36% gradient of B buffer (80% acetonitrile and 0.1% formic acid in H2O) for peptide separation, following with two steps washes: 2 min of 36–100% and 6 min of 100% B buffer) with an Easy-nLC 1200 connected online to a Fusion Lumos mass spectrometer (Thermo). Scans were collected in data-dependent top-speed mode with dynamic exclusion at 90 s. Raw data were analyzed using MaxQuant version 1.6.0.1 search against Mouse Fasta database, with label free quantification and match between runs functions enabled. The output protein group was analyzed and visualized using DEP package as described before[44].

**Statistical information.** Data are presented as mean ± s.d. as indicated in the figure legends. Unpaired two-tailed student $t$ test, The $P$ value was calculated with the Prism 6 software. A $P < 0.05$ was considered as statistically, $*P < 0.05$, $**P < 0.01$, $***P < 0.001$. No statistical method was used to predetermine sample size. The experiments were not randomized. The investigators were not blinded to allocation during experiment and outcome assessment.

**Reporting summary.** Further information on research design is available in the Nature Research Reporting Summary linked to this article.

## Data availability

The RNA-Seq, ChIP-seq data have been deposited in the Gene Expression Omnibus database under the accession code GSE135451. The mass spectrometry proteomics data have been deposited to the ProteomeXchange Consortium (http://proteomecentral.proteomexchange.org) via the iProX partner repository[45] with the dataset identifier PXD026208. Source data are provided with this paper. The source data underlying Figs. 1e, 2d, f, 3c, d, 4c, 5b, 6c, d and Supplementary Figs. 1d, g, 2b, d, f, 3c–e, 4c, e–g, 6e, 7b, c, f, g are provided as a Source Data file. Supplementary Data 1 and 2 are primers used in this study. The Cas9 screen data and IP-MS data are provided in Supplementary Data 3 and 4, respectively. The authors declare that all data supporting the findings of this study are available within the article and its supplementary information files or from the corresponding author upon reasonable request. Source data are provided with this paper.

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

## Acknowledgements

This work was supported in part by National Natural Science Foundation of China (31830060, 32022019, U20A2013, 31530038, 31770889, 31801180); the National Key R&D Program of China (2017YFA0504100, 2018YFE0204800, 2017YFA0105001, 2018YFA0108700) "Strategic Priority Research Program" of the Chinese Academy of Sciences (XDA16010505), Key Research Program of Frontier Sciences of the Chinese Academy of Sciences (QYZDJ-SSW-SMC009), Science and Technology Planning Project of Guangdong Province (2017B030314056, 2018B030306047) Guangzhou Regenerative Medicine and Health Guangdong Laboratory project (2018GZR110104003, 2018GZR110105012). We would like to thank Liman Guo from Metabolomics and Proteomics Center, Bioland Laboratory for proteomics measurements. The authors also are grateful for the support from the Guangzhou Branch of the Supercomputing Center of the Chinese Academy of Sciences. All the animal experiments were performed with the approval and according to the guidelines of the animal care and use committee of the Guangzhou Institutes of Biomedicine and Health.

## Author contributions

J.L. and J.K. designed the project; J.K. set up the CRISPR/cas9 screening system and phase separate analysis; J.K., C.W., S.Y., Z.Z., J.G., and W.L. performed the screening experiments, Z.Z., Z.C., W.L., Y.Q. and S.C. performed the RNA-seq experiments; W.G., Z.Z., performed FRAP experiments. P.L. and Z.Z. constructed the plasmids. T.Y. provided glial cells. J.K., C.B., Z.W., B.W., and X.L. performed the ChIP-seq experiments; R.S., D.L., and X.Z. performed the IP-MS experiments, J.K., Y.Y., S.X., J.H., J.C., and G.Z. analyzed the data; J.L. and D.P. supervised the whole study, conceived the whole study, wrote the paper, and approved the final version.

## Competing interests

The authors declare no competing interests.
