## [Peer Review File · Nature Communications]

REVIEWER COMMENTS

Reviewer #1

Remarks to the Author:

The new manuscript has been very much improved and criticised experiments are either properly/better controlled or removed. New experiments have been added making this a rather different manuscript.

I was very puzzled with respect to figure 6 and extended 6 and 7.

Figure 6B is claimed to show co-expression analysis.

The setup is not very specific because: lentiviral SS18-EGFP with antibodies and BAF or PBAF-specific antibodies detecting the respective endogenous proteins. My problem is that there seems to be a strong difference in the levels of SS18 and other components of the BAF complex (a few spots of SS18-EGFP and a strong nuclear staining by the BAF and PBAF specific antibodies. This even though lentiviral expression of SS18-EGFP is used which (normally) yields higher levels of the protein than endogenous.

This is at odds with the experiments up till this point which suggested that SS18 is a genuine subunit of the BAF complex.

From Figure 6 it would seem that SS18 is a substoichiometric subunit. This asks for several control experiments that validate not only the results of figure 6 but also some of the earlier figures.

Another point is how they got to the model in 6d and 6e. I truly do not know and have not seen experiments from which this can be concluded.

Reviewer #1 (Remarks to the Author):

The new manuscript has been very much improved and criticised experiments are either properly/better controlled or removed. New experiments have been added making this a rather different manuscript.

Response: We provided additional data to address the role of SS18 in PST, which was a concern of editor and the other reviewer.

I was very puzzled with respect to figure 6 and extended 6 and 7.

Response: We are sorry for not explaining figure 6 and extended 6 clearly, and we will explain in detail below. But, Extended 7 is uncropped western blot images organized as required.

Figure 6B is claimed to shows co-expression analysis.

Response: Figure 6B is co-localization analysis of SS18 and subunits of BAF or PBAF. We just overexpressed SS18-EGFP by lentivirus, instead of co-expression. We DID not claim this as co-expression, but co-localization analysis.

The setup is not very specific because: lentiviral SS18-EGFP with antibodies and BAF or PBAF-specific antibodies detecting the respective endogenous proteins. My problem is that there seems to be a strong difference in the levels of SS18 and other components of the BAF complex (a few spots of SS18-EGFP and a strong nuclear staining by the BAF and PBAF specific antibodies. This even though lentiviral expression of SS18-EGFP is used which (normally) yields higher levels of the protein than endogenous.

Response: Given the misunderstanding above as well as additional misunderstanding, the reviewer became discouraged about our last revision. We apologize for not being able to put forward a clearer narrative about the changes.

First misunderstandings were probably caused by select photography and image processing. Given the fact that condensates vary in intensity and the protein within condensates is extremely concentrated and much higher than the outside, especially for

the fused condensates (more than 50 times the fluorescence intensity outside the fused condensates in our experiments), regular photography and image processing tend to overexpose them. To solve this problem, we use very low laser intensity (2.5%) and detector gain (500V), as well as background subtraction treatment to obtain those images. Similar situations have been reported in other reported phase separation protein, like SH3(Fig. 3a¹), BRD3(Fig. 3d²), SRSF2(Figure. 2b³) and TDP-43(Figure. 1c⁴).

In this version, we compared different processed images (Extended Data Fig. 6b). To avoid overexposure of the condensates, we used low laser intensity (2.5%) and detector gain (500V), as well as background subtraction treatment as described above. On the contrary, if the images taken normally (the same parameters of photography as endogenous SS18 immunofluorescence in Extended Data Fig. 3a and endogenous SS18-EGFP this version of manuscript in Extended Data Fig. 6d, laser intensity 3.5%, detector gain 750V), we show that a large amount of SS18-EGFP are also present outside the few spots. We hope that this is clear to the reviewer. We can provide more details if needed.

This is at odds with the experiments up till this point which suggested that SS18 is a genuine subunit of the BAF complex.

From Figure 6 it would seem that SS18 is a substoichiometric subunit. This asks for several control experiments that validate not only the results of figure 6 but also some of the earlier figures.

Response: From the overexposed images above, we can see SS18 not only exists in the few spots (protein condensates), but also many more places in lower intensities. Consistently, the fluorescence in the condensates can recover quickly after bleaching in that a large amount of SS18 exists outside the spots (Fig. 3a).

We also knocked-in EGFP within SS18 loci to label it as SS18-EGFP for the endogenous pattern (Extended Data Fig. 6c). It's consistent with Figure 6a/b that the condensates formed by endogenous SS18 co-localize with DPF2(Extended Data Fig. 6d, e). The condensates formed by endogenous SS18-EGFP (Extended Data Fig. 6d) or nuclear staining of endogenous SS18 (Extended Data Fig. 3a) are comparable with that

of BAF specific component.

It's worth mentioning that a recent article on the analysis of endogenous mammalian BAF structure also confirmed that SS18 is a stoichiometric subunit of BAF, and SS18 and DPF2 are involved in BAF assembly at a ratio of 1:1⁵.

Another point is how they got to the model in 6d and 6e. I truly do not know and have not seen experiments from which this can be concluded.

Response : We are sorry for not explaining the model clearly and we will describe in more detail in the text and figure legends. We can also remove the model if this can cause confusion to the reviewer as well as the readers. This is to describe a balance between BAF and PBAF which share components such as BRG1. In the absence of SS18, more PBAF forms at the expense of BAF, resulting defects in PST. We have delineated this process into a 6 steps process. Again, this model is consistent with our data, but we can remove this if that would make the paper stronger.

We adjusted the direction of the model to make the scales look more vivid.

1 and 2 : BRG1, as a shared component, can be assembled into BAF and PBAF complexes⁶.

3: SS18 is a liquid-liquid phase separation protein (Fig. 3 and Extended Data Fig. 3) whose condensates can co-localize with DPF2 (Fig. 6a and Extended Data Fig. 6d, e)

4: The condensates formed by SS18 isolate BRD7 (Fig. 6b and Extended Data Fig. 6d, e).

5: By immunoprecipitating BRG1 in SS18 KO mESCs and those with restoring expression of SS18 N-terminal (SS18 N-terminal itself can interact with BAF, showed in Fig. 5c) or SS18 full length (Extended Data Fig. 6f), we found that the amount relative to BRG1 of BAF-specific components increased while the PBAF-specific ones decreased upon SS18 recovering (Fig. 6c). In other words, SS18 can affect the balance of different complexes amounts by facilitating the assembly of BAF and impair that of PBAF.

6: The knockdown experiments showed that it's BAF that serves to promote cJUN induced PST, instead of PBAF (Fig. 6d). Consequently, the reduced amount of BAF due to SS18 deficiency will delay the PST process.

In summary, we speculate SS18, interacting with BAF subunits and forming condensates, mediates the assembly balance between BAF and PBAF, and regulates the cJUN induced PST process.

Reference:

1. Li, P.L. *et al.* Phase transitions in the assembly of multivalent signalling proteins. *Nature* **483**, 336-U129 (2012).
2. Daneshvar, K. *et al.* lncRNA DIGIT and BRD3 protein form phase-separated condensates to regulate endoderm differentiation. *Nature Cell Biology* **22**, 1211-+ (2020).
3. Guo, Y.E. *et al.* Pol II phosphorylation regulates a switch between transcriptional and splicing condensates. *Nature* **572**, 543-+ (2019).
4. Gasset-Rosa, F. *et al.* Cytoplasmic TDP-43 De-mixing Independent of Stress Granules Drives Inhibition of Nuclear Import, Loss of Nuclear TDP-43, and Cell Death. *Neuron* **102**, 339-+ (2019).
5. Mashtalir, N. *et al.* A Structural Model of the Endogenous Human BAF Complex Informs Disease Mechanisms. *Cell* **183**, 802-817 e824 (2020).
6. Ho, L. & Crabtree, G.R. Chromatin remodelling during development. *Nature* **463**, 474-484 (2010).

REVIEWERS' COMMENTS

Reviewer #4 (Remarks to the Author):

Kuang and colleagues use a model system of cJUN-induced pluripotent to somatic transition (PST) to identify SS18 as a critical regulator of PST using a CRISPR based genome wide screen. They validate these results by generating an SS18 KO mouse embryonic stem cell line, which has increased propensity to maintain the pluripotent state in the presence of cJUN expression as measured by OCT4-GFP fluorescence and differential gene expression analysis. The authors demonstrate that SS18 can form condensates and identify an IDR region in SS18 that regulates condensate formation, as well as an N terminal region that regulates binding to BAF complexes. The N terminal region appears to be required for rescue of the PST, suggesting SS18 is working through BAF complexes to mediate this effect. The IDR is also required, but function can be conferred by the Fus IDR or PolyY IDR. Finally, they show that SS18 overexpression in SS18KO mESCs results in increased association of BRG1 with DPF2, ARID1A, and ARID1B and decreased association with PBAF components PBRM1, BRD7, ARID2, and PHF10. They further show that while BAF complex subunits are required for PST, knock down of PBAF components either have no effect or enhance PST.

Overall, this is an interesting study with new insights into the action of SS18 in developmental transitions. However, I have reservations about the interpretations regarding the assembly of BAF and PBAF variants and the lack of investigation of the ncBAF complex.

- 1) The experiments showing that SS18 promotes assembly of the BAF complex were done not by comparing SS18-WT and SS18-KO, but by comparing SS18 overexpression against a GFP vector control in SS18-KO mESCs. The model in Figure 6e should be adjusted to reflect the data and show a shift in the balance to BAF complexes in the overexpression system rather than a shift to PBAF complexes in the SS18-KO, which was not investigated. Because this shift could be caused by overexpression of the SS18 protein, it is wrong to assume the SS18-KO phenotype results in an enrichment of PBAF complexes. Further, the statement "Taken together, these data suggest that SS18 promotes BAF but inhibits PBAF assembly to mediate PST (Fig.6e)." should be edited to remove "inhibits PBAF" as this was not shown. Alternatively, the authors can perform SS18-WT and SS18-KO comparisons as for Figure 6c to establish this point.
- 2) The authors seem to have missed a major advance in the field by not acknowledging the existence of new variant of the complex, referred to as the GBAF or ncBAF complex (Alpsoy JCB 2018, Michel NCB 2018, Gatchalian Nat Comm 2018, Mashtalir Cell 2018 etc), which uniquely incorporates BRD9, GLTSCR1/1L. Further, Gatchalian Nat Comm 2018 demonstrates the role of BRD9 and this complex in maintaining naïve pluripotency during the naïve-primed transition, which the authors also explore in SS18-KO mESCs. The inference from these data would be that Brd9, Gltscr1, and/or Gltscr1 knockdown would behave similar to Brd7 and PBAF subunit knockdown to have no effect on Nanog expression or reduce Nanog expression further during PST. The authors should complete knockdown of these components to add to Figure 6d. BRD9 degrader (dBRD9) or inhibitors (BI-9564 or I-BRD9) could also be used in this assay. Further, the statement, "mESCs have two types of BAF complexes that share many core subunits such as BRG1, SMARCD1, SMARCB1, SMARCE1 and BCL7A/C, but differ in specific ones, i.e., the classic BAF with DPF1, DPF2, ARID1A, ARID1B and SS18 while the PBAF (Polybromo-associated BAF) with PBRM1, PHF10, ARID2 and BRD7 (Extended Data Fig.6a)." is incorrect and should be edited to include the third form of the complex and references.
- 3) A related point is whether PBAF or ncBAF complexes promote the pluripotent state and the expression of Nanog in the absence of SS18 in the c-JUN TetON setting, which the authors could test by knocking down components of either of these complexes in the SS18-KO or SS18 shRNA lines to see rescue of PST progression.
- 4) Interestingly, SS18 is associated with the ncBAF and BAF complex, but not the PBAF complex.

SMARCB1 on the other hand, is specific for the BAF and PBAF complexes. The authors should analyze the data from the experiments in Figure 6c (no new experiments required) to determine whether the N-terminal portion or the FL overexpression of SS18 promote the assembly of the ncBAF complex by looking at the abundance of BRD9, GLTSCR1, and GLTSCR1L associated with BRG1. SS18 overexpression could drive the assembly of the ncBAF complex even if the ncBAF complex not required for the PST.

5) To complete the work-up of the three variants, BRD9 ChIP-seq could be performed and compared to the profiles in Extended Data Figure 6j, although this would not be absolutely necessary.

Minor points:

1) Along with Kidder et al, the authors could include Ho et al. PNAS 2009 PMID: 19279220 and PMID: 19279218

Reviewer #5 (Remarks to the Author):

In this manuscript, Kuang et al. first performed a genome-scale CRISPR screen to identify genes required for exit from pluripotency in their c-Jun-based differentiation system and identified Ss18 as a top hit. Through the follow-up analysis, they found that Ss18 forms condensates in the nucleus using the C-terminal intrinsically disordered region and facilitates BAF complex formation mediated by interaction through the N-terminal domain. BAF then promotes exit from pluripotency.

The manuscript is clearly written and well structured. The data are presented in a well-organized fashion.

Major comments:

1. The CRISPR screening strategy seems mostly appropriate. However, the comparison between Output and Input2 may increase false positives, depending on the duration between the two harvests. There are genes that when disrupted increase proliferation rates. These genes may also be identified from this comparison. The best control sample for this screen would have been dox-untreated cells cultured for the same period of time. In addition, 24-h dox treatment might not be the best treatment duration for this screen. If cell survival after dox treatment in the actual screening experiment is similar to that shown in Supplementary Figure 1D, the library complexity will be severely affected and statistical robustness will therefore be diminished. According to the read count provided, it seems that this is partially the case. Nevertheless, as Ss18 was identified in the two independent screens and validated in individual experiments, it would be reasonable to further study its function as a screen hit.

2. CRISPR screening has already been employed to identify genes required for exit from pluripotency (PMID: 29996108). Together with other genetic screens, there are gene consistently identified, such as Tcf7l1. Why could the authors not identified previously characterised genes in their system? Would the c-Jun-mediated exit be highly artificial? This seems partially addressed in Supplementary Figure 2e-g, but this result do not fully explain the discrepancy between the previous genetic screens and the current study. The authors should discuss this point.

3. I doubt if 'SS18 is a critical regulator at the pluripotent-somatic interface'. The definition of 'critical' is an arguable point, but it seems that SS18 is required to execute exit from pluripotency in a timely manner, but not critical for the exit as SS18 KO cells seem to be able to downregulate naïve-related genes, hence differentiate. Furthermore, the results presented in Figure 6 seem to suggest that

functional BAF complex can form without SS18, although SS18 affect the balance between BAF and PBAF. As for the title, it would be more appropriate to state otherwise.

REVIEWERS' COMMENTS

Reviewer #4 (Remarks to the Author):

Kuang and colleagues use a model system of cJUN-induced pluripotent to somatic transition (PST) to identify SS18 as a critical regulator of PST using a CRISPR based genome wide screen. They validate these results by generating an SS18 KO mouse embryonic stem cell line, which has increased propensity to maintain the pluripotent state in the presence of cJUN expression as measured by OCT4-GFP fluorescence and differential gene expression analysis. The authors demonstrate that SS18 can form condensates and identify an IDR region in SS18 that regulates condensate formation, as well as an N terminal region that regulates binding to BAF complexes. The N terminal region appears to be required for rescue of the PST, suggesting SS18 is working through BAF complexes to mediate this effect. The IDR is also required, but function can be conferred by the Fus IDR or PolyY IDR. Finally, they show that SS18 overexpression in SS18KO mESCs results in increased association of BRG1 with DPF2, ARID1A, and ARID1B and decreased association with PBAF components PBRM1, BRD7, ARID2, and PHF10. They further show that while BAF complex subunits are required for PST, knock down of PBAF components either have no effect or enhance PST.

Overall, this is an interesting study with new insights into the action of SS18 in developmental transitions. However, I have reservations about the interpretations regarding the assembly of BAF and PBAF variants and the lack of investigation of the ncBAF complex.

1) The experiments showing that SS18 promotes assembly of the BAF complex were done not by comparing SS18-WT and SS18-KO, but by comparing SS18 overexpression against a GFP vector control in SS18-KO mESCs. The model in Figure 6e should be adjusted to reflect the data and show a shift in the balance to BAF complexes in the overexpression system rather than a shift to PBAF complexes in the SS18-KO, which was not investigated. Because this shift could be caused by overexpression of the SS18 protein, it is wrong to assume the SS18-KO phenotype results in an enrichment of PBAF complexes. Further, the statement "Taken together, these data suggest that SS18 promotes BAF but inhibits PBAF assembly to mediate PST (Fig.6e)." should be edited to remove "inhibits PBAF" as this was not shown. Alternatively, the authors can perform SS18-WT and SS18-KO comparisons as for Figure 6c to establish this point.

Response: We appreciate these suggestions and have performed IP-MS on SS18-WT and SS18-KO. The resulting dataset is now in the new Figure 6c. We found ncBAF behaves like canonical BAF in SS18 KO.

2) The authors seem to have missed a major advance in the field by not acknowledging the existence of new variant of the complex, referred to as the GBAF or ncBAF complex (Alpsoy JCB 2018, Michel NCB 2018, Gatchalian Nat Comm 2018, Mashtalir Cell 2018 etc), which uniquely incorporates BRD9, GLTSCR1/1L. Further, Gatchalian Nat Comm 2018 demonstrates the role of BRD9 and this complex in maintaining naïve pluripotency during the naïve-primed transition, which the authors also explore in SS18-KO mESCs. The inference from these data would be that Brd9, Gltscr1, and/or Gltscr1l knockdown would behave similar to Brd7 and PBAF subunit knockdown to have no effect on Nanog expression or reduce Nanog expression further during PST. The authors should complete knockdown of these components to add to Figure 6d. BRD9 degrader (dBRD9) or inhibitors (BI-9564 or I-BRD9) could also be used in this assay. Further, the statement, "mESCs have two types of BAF complexes that share many core subunits such as BRG1, SMARCD1, SMARCB1, SMARCE1 and BCL7A/C, but differ in specific ones, i.e., the classic BAF with DPF1, DPF2, ARID1A, ARID1B and SS18 while the PBAF (Polybromo-associated BAF) with PBRM1, PHF10, ARID2 and BRD7 (Extended Data Fig.6a)." is incorrect and should be edited to include the third form of the complex and references.

Response: We are sorry for missing this advance and added the data about ncBAF in this version as suggested. Interestingly, ncBAF plays a role similar to canonical BAF (Figure 6d). However, inhibitor BI-9564 does not have any effect during the process (Supplementary figure 7f). We have revised the statement mentioned by the reviewer.

3) A related point is whether PBAF or ncBAF complexes promote the pluripotent state and the expression of Nanog in the absence of SS18 in the c-JUN TetON setting, which the authors could test by knocking down components of either of these complexes in the SS18-KO or SS18 shRNA lines to see rescue of PST progression.

Response: We appreciate these suggestions. We show that knocking down components of ncBAF leads to a further delay in PST, and the knockdown of PBAF rescues that (Supplementary figure 7g), suggesting that BAF and ncBAF play similar/redundant roles while PBAF plays an opposite role in cJUN induced PST.

4) Interestingly, SS18 is associated with the ncBAF and BAF complex, but not the PBAF complex. SMARCB1 on the other hand, is specific for the BAF and PBAF complexes. The authors should analyze the data from the experiments in Figure 6c (no new experiments required) to determine whether the N-terminal portion or the FL overexpression of SS18 promote the assembly of the ncBAF complex by looking at the abundance of BRD9, GLTSCR1, and GLTSCR1L associated with BRG1. SS18 overexpression could drive the assembly of the ncBAF complex even if the ncBAF complex not required for the PST.

Response: We appreciate these suggestions. By comparing SS18-WT and SS18-KO, we show that ncBAF specific components decrease without SS18 (figure 6c). We also show that SS18 FL, not the N terminal down, can rescue ncBAF assembly (Supplementary figure 7b). In addition, we show now that ncBAF components participate in PST (figure 6d).

5) To complete the work-up of the three variants, BRD9 ChIP-seq could be performed and compared to the profiles in Extended Data Figure 6j, although this would not be absolutely necessary.

Response: We agree with the suggestions of the reviewer and indeed it's not absolutely necessary in this manuscript. We may perform this in the near future and report that when appropriate.

Minor points:

1) Along with Kidder et al, the authors could include Ho et al. PNAS 2009 PMID: 19279220 and PMID: 19279218

Response: We included that as suggested.

Reviewer #5 (Remarks to the Author):

In this manuscript, Kuang et al. first performed a genome-scale CRISPR screen to identify genes required for exit from pluripotency in their c-Jun-based differentiation system and identified Ss18 as a top hit. Through the follow-up analysis, they found that Ss18 forms condensates in the nucleus using the C-terminal intrinsically disordered region and facilitates BAF complex formation mediated by interaction through the N-terminal domain. BAF then promotes exit from pluripotency.

The manuscript is clearly written and well structured. The data are presented in a well-organized fashion.

Major comments:

1. The CRISPR screening strategy seems mostly appropriate. However, the comparison between Output and Input2 may increase false positives, depending on the duration

between the two harvests. There are genes that when disrupted increase proliferation rates. These genes may also be identified from this comparison. The best control sample for this screen would have been dox-untreated cells cultured for the same period of time.

Response: We really appreciated these kind suggestions. In rep2 screen, we did collect the sample without dox treatment (named input3) at the same time when collecting output sample as suggested by the reviewer. The top hits from output_vs_input2 and output_vs_input3 are similar, 9 out of top 10 hits and 17 out of top 20 hits are overlapped. In our screening pipeline, in addition to the sgRNAs which can delay PST process, sgRNAs influencing cell survival or apoptosis may also be enriched, thus, contributing to some hits failed in our PST validation (by Nanog expression level). In the top hits, like Myl6 Ric8 Rhoa Rac1 and Myh9, are all involved in ROCK-dependent apoptosis pathway (Ohgushi et al., 2010; Zhao et al., 2015). Interestingly, P53 may be involved in both cell survival and differentiation, also showed up in our screen.

Reference:

Ohgushi, M., Matsumura, M., Eiraku, M., Murakami, K., Aramaki, T., Nishiyama, A., Muguruma, K., Nakano, T., Suga, H., Ueno, M., et al. (2010). Molecular pathway and cell state responsible for dissociation-induced apoptosis in human pluripotent stem cells. *Cell Stem Cell* 7, 225-239.

Zhao, B., Qi, Z., Li, Y., Wang, C., Fu, W., and Chen, Y.G. (2015). The non-muscle-myosin-II heavy chain Myh9 mediates colitis-induced epithelium injury by restricting Lgr5+ stem cells. *Nat Commun* 6, 7166.

In addition, 24-h dox treatment might not be the best treatment duration for this screen. If cell survival after dox treatment in the actual screening experiment is similar to that shown in Supplementary Figure 1D, the library complexity will be severely affected and statistical robustness will therefore be diminished. According to the read count provided, it seems that this is partially the case. Nevertheless, as Ss18 was identified in the two independent screens and validated in individual experiments, it would be reasonable to further study its function as a screen hit.

Response: We appreciate the in-depth comments by the reviewer. We also felt that the second screen was crucial as both identified SS18, the focus of our report.

2. CRISPR screening has already been employed to identify genes required for exit from pluripotency (PMID: 29996108). Together with other genetic screens, there are gene consistently identified, such as Tcf711. Why could the authors not identified previously characterised genes in their system? Would the c-Jun-mediated exit be highly artificial? This seems partially addressed in Supplementary Figure 2e-g, but this result do not fully explain the discrepancy between the previous genetic screens and the current study. The authors should discuss this point.

Response: We appreciate the reviewer's comments. The previously reported pluripotency exit models are mostly about changes of culture conditions, like 2i withdrawal. On the contrary, our cJUN induced exit from pluripotency is a different approach by design. Actually, we also performed 2i withdrawal screen along with the cJUN induced one and in deed, Tcf711 is one of the top hits (data not shown).

3. I doubt if 'SS18 is a critical regulator at the pluripotent-somatic interface'. The definition of 'critical' is an arguable point, but it seems that SS18 is required to execute exit from pluripotency in a timely manner, but not critical for the exit as SS18 KO cells seem to be able to downregulate naïve-related genes, hence differentiate. Furthermore, the results presented in Figure 6 seem to suggest that functional BAF complex can form without

SS18, although SS18 affect the balance between BAF and PBAF. As for the title, it would be more appropriate to state otherwise.

Response: We used it as it was suggested by an earlier reviewer in NCB. Now we have changed it as suggested. We are open to any suggestion at this point.